# First Insights on Soil Respiration Prediction across the Growth Stages of Rainfed Barley Based on Simulated MODIS and Sentinel-2 Spectral Indices

**Víctor Cicuéndez [1], Manuel Rodríguez-Rastrero [2], Laura Recuero [1,3], Margarita Huesca [4], Thomas Schmid [2], Rosa Inclán [2], Javier Litago [5], Víctor Sánchez-Girón [6] and Alicia Palacios-Orueta [1,3,*]**

[1]   Departamento de Sistemas y Recursos Naturales, ETSIMFMN, Universidad Politécnica de Madrid (UPM), 28040 Madrid, Spain; victor.cicuendez.lopezocana@alumnos.upm.es (V.C.); laura.recuero@upm.es (L.R.)

[2]   Centro de Investigaciones Energéticas, Medioambientales y Tecnológicas (CIEMAT), 28040 Madrid, Spain; manuel.rodriguezrastrero@ciemat.es (M.R.-R.); thomas.schmid@ciemat.es (T.S.); rm.inclan@ciemat.es (R.I.)

[3]   Centro de Estudios e Investigación para la Gestión de Riesgos Agrarios y Medioambientales (CEIGRAM), Universidad Politécnica de Madrid (UPM), 28040 Madrid, Spain

[4]   Center for Spatial Technologies and Remote Sensing (CSTARS) of the Land, Air and Water Resources Department, University of California Davis, CA 95616, USA; mhuescamartinez@ucdavis.edu

[5]   Departamento de Economía Agraria, Estadística y Gestión de Empresas, ETSIAAB, Universidad Politécnica de Madrid (UPM), 28040 Madrid, Spain; javier.litago@upm.es

[6]   Departamento de Ingeniería Agroforestal, ETSIAAB, Universidad Politécnica de Madrid (UPM), 28040 Madrid, Spain; victor.sanchezgiron@upm.es

*   Correspondence: alicia.palacios@upm.es

**Abstract:** Rainfed agriculture occupies the majority of the world's agricultural surface and is expected to increase in the near future causing serious effects on carbon cycle dynamics in the context of climate change. Carbon cycle across several temporal and spatial scales could be studied through spectral indices because they are related to vegetation structure and functioning and hence with carbon fluxes, among them soil respiration (Rs). The aim of this work was to assess Rs linked to crop phenology of a rainfed barley crop throughout two seasons based on spectral indices calculated from field spectroscopy data. The relationships between Rs, Leaf Area Index (LAI) and spectral indices were assessed by linear regression models with the adjusted coefficient of determination ($R_{adj}^2$). Results showed that most of the spectral indices provided better information than LAI throughout the studied period and that soil moisture and temperature were relevant variables in specific periods. During vegetative stages, indices based on the visible (VIS) region showed the best relationship with Rs. On the other hand, during reproductive stages indices containing the near infrared-shortwave infrared (NIR-SWIR) spectral region and those related to water content showed the highest relationship. The inter-annual variability found in Mediterranean regions was also observed in the estimated ratio of carbon emission to carbon fixation between years. Our results show the potential capability of spectral information to assess soil respiration linked to crop phenology across several temporal and spatial scales. These results can be used as a basis for the utilization of other remote information derived from satellites or airborne sensors to monitor crop carbon balances.

**Keywords:** phenology; leaf area index; carbon balance; field spectroscopy; Mediterranean climate

## 1. Introduction

Agricultural areas represent 11% of the global land surface [1] generating around 13% of the greenhouse gases emissions and are expected to increase in the near future [2,3]. Soil $CO_2$ efflux, termed also as soil respiration (Rs), accounts for 60–80% of the total ecosystem respiration [4,5] being the main way by which $CO_2$ fixed by land vegetation returns to the atmosphere [6,7]. Soil autotrophic (Ra) and heterotrophic respiration (Rh) are highly dependent on biotic and abiotic factors. Heterotrophic respiration coming from the organic matter mineralization by the decomposer community is highly affected by the availability of soil carbon, moisture and temperature [8–10]. The importance of the autotrophic component has been extensively demonstrated [11–13]. Ra rate patterns are highly related to photosynthesis [14–16] and therefore strongly mediated by phenological dynamics along crop cycles [17,18]. In addition, the accumulation of carbon exudates in roots as well as variation in canopy transpiration and shading through the phenological cycle creates local soil conditions modifying heterotrophic respiration processes [19]. In agricultural fields all these factors result in a high $CO_2$ fluxes spatial and temporal variability [20,21] which may be stronger in rainfed crops in Mediterranean climates due to the irregular distribution of rainfall and temperature [22,23].

Chamber-based methods are commonly used to measure Rs [24], however, measurements are local so that do not account for the Rs spatial and temporal variability at ecosystem scale [25]. Temporal frequency of the measurements has been improved by the use of techniques such automated soil respiration chambers [25] and soil gradient methods [26].However, these methodologies remain providing measurements at plot level making necessary to carry out a large number of them to obtain a reliable representation of spatial variability [27–29]. To estimate Rs at larger scales some authors developed models based on Rs from several data bases [30,31]. However, there is a need to develop methods for estimating soil respiration through the growing period on a frequent basis and at large scales [28,32,33].

Remote sensing data is the most appropriate tool for monitoring crop's phenology across several temporal and spatial scales [34,35]; the link between spectral Indices and physiological processes [36,37] opens new ways to assess soil respiration patterns at large scale.

Huang et al. [38] developed linear regressions models using the Normalized Difference Vegetation Index (NDVI) [39], the Enhanced Vegetation Index (EVI) [40] and the Red Edge Chlorophyll Index ($Cl_{red\,edge}$) [41] from spectral field measurements in irrigated maize and winter wheat crops. Also in irrigated maize, Huang et al. [42] found the best model to estimate Rs when including Soil Organic Content (SOC) together with satellite EVI data at large scale in an exponential model. In irrigated crops, Huang et al. [33] used EVI from Landsat 8, SOC and bulk density to upscale plot soil respiration, finding better results when using a support vector regression model than a multiple regression model. In the last two studies, however, the models were only appropriate for the central period of the growing season and may not provide reliable estimations in other crop stages.

Based on satellite data only, Huang et al. [43] modeled Rs at large scale in alpine grasslands through the peak growing season using NDVI, EVI and Modified Soil Adjusted Vegetation Index (MSAVI) [44] from MODIS and Landsat data. They found the best estimation when using an NDVI exponential model, due probably to the high proportion of the autotrophic component in total Rs during this period. In cold temperate coniferous sites, Yan et al. [45] developed a model based on MODIS land surface temperature as the primary factor, however, they found that the Rs estimation accuracy was improved when including NDVI.

Cicuéndez et al. [46] included soil moisture and temperature together with field spectroscopy vegetation Indices in multiple linear regression to estimate Rs in an irrigated maize crop. Temperature improved the Rs estimation through all the growing period, while including moisture did not show significant differences. In this work a first assessment of Ra estimation showed significant results during the vegetative stages. Huang and Niu [47] also found a significant influence of abiotic factor to estimate Rs in a rainfed crop with an irregular Rs pattern.

Spectral Shape Indices (SSI) [48] use information of three consecutive bands, taking advantage of spectral shape in different spectral ranges and facilitating the connection between two spectral regions. SSI in the SWIR have been used to monitor crop phenological patterns including periods when photosynthetic activity is not dominant [35,49] as well as in agricultural drought assessment [50].

This concept has also been applied to estimating residue biomass from crops [51,52], to detect spruce bark beetle infections [53] and for non-destructive estimations of foliar carotenoid content of tree species [54] among others. In addition, Cicuéndez et al. [46] have shown that SSI could also be used in Rs assessment in a maize crop.

Most of the studies estimating Rs in agricultural fields have been accomplished in irrigated crops. However, rainfed crops represent 80% of the cultivated land surface and 60% of the agricultural production [55] with a high inter-annual variability, hence there is an urgent need to assess Rs in this type of agriculture at large scale. Specifically, barley (*Hordeum vulgare* ssp. *Vulgare* L.) is the fourth most widely grown crop worldwide and the first rainfed crop in extension and production in Spain [56].

The general goal of this work was to assess the capability of several spectral indices, including SSI, based on field hyperspectral information, to assess soil respiration (Rs) linked to crop phenology in a rainfed barley crop in Mediterranean climate. Specific objectives were the following: (1) assessing differences between spectral indices and Leaf Area Index (LAI) to model Rs during vegetative and reproductive stages, (2) to explore the role of soil temperature and moisture on Rs assessment and (3) to carry out a first approximation to assess carbon emission vs. carbon fixation based on the relationship between Rs and barley biomass.

## 2. Materials and Methods

### 2.1. Experimental Field

The study was carried out in a rainfed barley field during two crop seasons (2012 and 2013), which is situatedat an altitude approximately of 600 m a.s.l. in the UTM zone 30N, 437410 East and 4476875 North (Datum ETRS89), corresponding to the experimental fields of the Technical School of Agricultural Engineering of the Universidad Politécnica de Madrid (Spain). It is a flat areawhich has been cultivated systematically for 30 years with different crops, including barley. The whole field is characterizedby homogeneous soil, slope, drainage and lighting [46]. The experimental area results from a filling of anthropic origin ("human-transported materials", [57]) consisting of arkosic materials with an abundance of artifacts. For such reasons, this soil belongs to the Xerorthents Group according to US Soil Taxonomy System [58], andit can be classified as Anthroportic Xerorthent according to Capra et al. [59]. Considering the barley main rooting zone (0–30 cm), soils are well-drained, basic (pH from 7.5 to 8.7), slightly saline (electrical conductivity from 2.5 to 3.1 dS m$^{-1}$) with a sandy-clay texture and organic carbon content of around 1%.

The area has a Mediterranean continental climate (Csa according to Köppen-Geiger classification) [60]. During the barley growing cycle, average temperatures range from 8.0 °C (March) to 20.1 °C (June) [61]. Rainfall slightly exceeds evapotranspiration during March and April, but since May evapotranspiration is higher than rainfall, defining a remarkable water deficit in rainfed conditions even before the full maturity of barley in June [61].

The field experimental design consisted in three plots with the same soil properties and management practices: two cultivated plots, identified as BY1 and BY2 and a control plot with bare soil (S) (Figure 1). The labels "12" and "13" are referred to the years 2012 and 2013 respectively. The total crop area was 1278 m$^2$ (71 m long × 18 m wide).

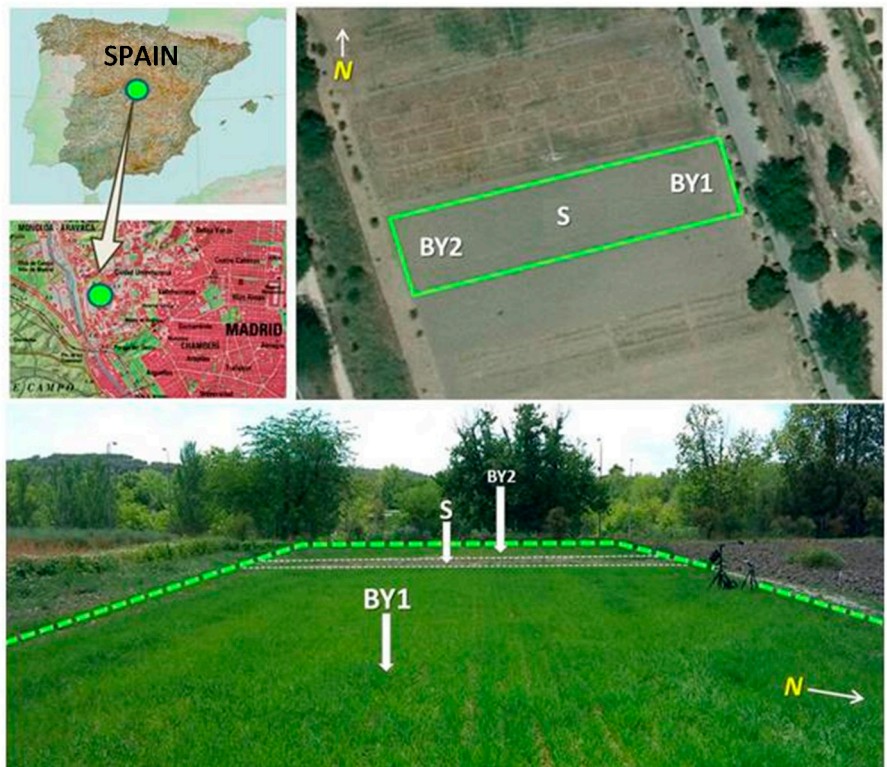

**Figure 1.** Experimental area and plots location (maps and orthoimage from IGN, www.ign.es/ign_iberpix/) and overview picture (13 May 2013).

### 2.2. Barley Phenology

In this study, a two-row spring-barley variety "UNIA-R1" was used. The phenological evolution of barley was assessed from March to July during 2012 and 2013. Phenological stages of barley were visually identified and marked according to Zadoks's cereal growth stages [62]. Based on the results obtained by Cicuéndez et al. [46], we have used the Zadoks's division in this work with models applied to vegetative and reproductive stages. The Day of Year (DOY) of the stages for 2012 and 2013 is shown in Table 1.

**Table 1.** Day of year (DOY) of vegetative and reproductive stages based on Zadoks's scale [62].

|  | Growth Stage | DOY 2012 | DOY 2013 |
|---|---|---|---|
| **Vegetative stages** | S | 70 | 81 |
|  | SG (10) | 82 | 98 |
|  | SG (12) | 88 | 102 |
|  | SG (13) | - | 108 |
|  | T1 | 100 | 113 |
|  | T2 | 108 | 123 |
|  | SE | - | 133 |
|  | B | 130 | 143 |
| **Flowering and reproductive stages** | F | 139 | 155 |
|  | D | 145 | 171 |
|  | R | 166 | 184 |
|  | H | 205 | 198 |

S: Sowing; SG (10): Seedling growth, first leaf through coleoptiles; SG (12): Seedling growth, two leaves emerged; SG (13): Seedling growth, three leaves emerged; T1: Tillering, first stages of tillering; T2: Tillering, last stages of tillering; SE: Stem elongation; B: Booting; F: Flowering, inflorescence emergence; D: Dough development; R: Ripening; H: Harvest.

Barley was sown in March in both years, specifically the 10 March 2012 (DOY 70) and the 22 March 2013 (DOY 81). Barley completed its cycle at the end of June or at the beginning of July, and it was completely ripened in July. The cycle from sowing to ripening was shorter in 2012 than in 2013, it lasted 96 days in 2012 and 103 days in 2013. The harvest was carried out in mid-July in both years, a few days after the last spectro-radiometric measurement.

### 2.3. Cultural Practices

The same experimental layout was used in both growing seasons. In all three plots moldboard plowing was performed as primary tillage (25–30 cm depth). Later, seedbed was prepared by one disk harrowing (10 cm depth), incorporating rabbit manure (50–60 Mg ha$^{-1}$) to the soil. According to Li-Li et al. [63], rabbit manure constitutes a remarkable source of N, P and K, including 1.8%, 0.6% and 0.7% respectively of such nutrients from dry matter. Then barley UNIA-R1 was sown in BY plots at a rate of 120 kg·ha$^{-1}$ with a seed drill with fluted coulters and double disk furrow openers. Glyphosate was applied to the bare soil area in some phases during the growing period although the eventual presence of weeds was usually manually controlled during both growing seasons.

### 2.4. Measurement of Soil $CO_2$ Efflux

Measurements of soil $CO_2$ efflux, expressed as Rs (μmols $CO_2$·m$^{-2}$·s$^{-1}$), were taken by a portable automated soil $CO_2$ infrared gas analyzer (Li-8100, Li-COR Inc., Lincoln, NE, USA) equipped with a 10 cm survey chamber (Model 8100-102) over PVC collars (105 mm of diameter and 90 mm high) which were placed at a depth of 5–6 cm in the soil and they were cleared of weeds during the growing season.

Five collars were distributed in each plot each year in a semicircular distribution near to spectro-radiometric measurement points (Figure 2). In each collar, an average of three measurements of $CO_2$ efflux was done and then the average of the five collars readings was calculated. As result, one measurement of Rs for each measurement day was estimated in each plot. Collars were located in similar places each year. All Rs measurements were taken between 11 pm and 14 pm.

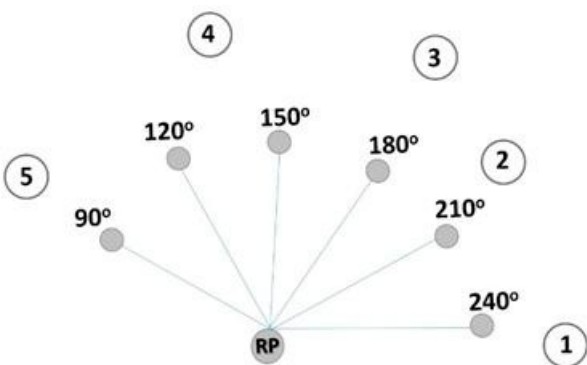

**Figure 2.** Scheme of the spectro-radiometric measurements (six angles from a reference point "RP") and soil respiration collars distribution (5 circles) in the cultivated plots.

Total $CO_2$ emissions each year were estimated by integrating the Rs values throughout the growing cycle.

### 2.5. Field Spectroscopy

An ASD FieldSpec3 Spectroradiometer (Analytical Spectral Devices Inc., Boulder, CO, USA) was used to take hyperspectral ground measurements within a spectral range from the visible (VIS) to Shortwave Infrared (SWIR) region (350–2500 nm), approximately each 15 days, selecting days with clear sky conditions. All measurements were taken between 12 pm and 4 pm.

One reference point was established in each plot for each year. Then, six points were selected at a distance of 1.3 m from the reference point in the cultivated plots according to six angles (90°,

120°, 150°, 180°, 210°, 240°) (Figure 2). In the bare soil plot, three points were selected at a distance of 1.3 m and three points were selected at a distance of 1.8 m. Measurements were made by placing the foreoptic (pistol grip) fixed on a tripod one meter above each of the resulting points.

Three spectro-radiometric measurements were obtained over plant canopy (BY) and bare soil (S) in each point (44cm field of view). Spectroradiometer was calibrated between points using a Spectralon reference panel (ASDInc., Boulder, CO, USA) and measurements were corrected for sensor discrepancy using a splice correction (ViewSpecPro5.6, ASD Inc., Boulder, CO, USA). Then, the average value of the three measurements taken in the six BY and S points was considered. After that, a Savitzky-Golay filter was applied using a second order polynomial with a window size of five spectral bands within the visible and near infrared spectra [46]. Then, a spectral resampling method to MODIS and to Sentinel-2 bands with the function spectral resampling of the software ENVI 4.2 (Research Systems Inc., Boulder, CO, USA) was done. In both cases seven bands in the optical domain were selected. Spectral indices were calculated from these bands (Table 2) and the average of six measurements was computed.

**Table 2.** Spectral indices obtained based on MODIS and Sentinel-2 wavelengths.

| Index | Equation | Description | Reference |
|---|---|---|---|
| NDVI | $\frac{(\rho_{nir}-\rho_{red})}{(\rho_{nir}+\rho_{red})}$ | Ratio Index based on the chlorophyll absorption at red band related to photosynthesis, vegetation growth and activity | [39] |
| EVI | $\frac{(\rho_{nir}-\rho_{red})}{(\rho_{nir}+C1*\rho_{red}-C2*\rho_{blue}+1)}$ | Ratio Index developed to optimize the vegetation signal, especially in high biomass regions | [40] |
| NDWI * | $\frac{(\rho_{nir}-\rho_{swir1})}{(\rho_{nir}+\rho_{swir1})}$ | Ratio Index for detecting water status, the reflectance properties of green vegetation, dry vegetation and soils | [64] |
| AG | Angle at green ($\alpha_{green}$) | SSI that takes into account three consecutive bands: blue, green and red. Related to pigment content and photosynthesis | [48] |
| AR | Angle at red ($\alpha_{red}$) | SSI that takes into account three consecutive bands: green, red and NIR Related to pigment content and photosynthesis | [48] |
| ANIR (only for MODIS) | Angle at NIR ($\alpha_{nir}$) | SSI that takes into account three consecutive bands: red, NIR and SWIR1. Related to photosynthetic capacity and moisture and is able to detect dry plant matter in the presence of soil and green vegetation | [52] |
| SASI (only for MODIS) | Angle at SWIR1 ($\alpha_{swir1} * Slope$) | SSI that takes into account three consecutive bands: NIR, SWIR1 and SWIR2. Related to moisture and non-photosynthetic activity, it is able to discriminate dry plant matter in the presence of soil and green vegetation | [52] |

$\rho_{blue}$: blue spectral band in MODIS and band 2 in Sentinel-2; $\rho_{red}$: red spectral band in MODIS and band 4 in Sentinel-2; $\rho_{nir}$: near-infrared spectral band in MODIS and band 8 in Sentinel-2; $\rho_{swir1}$: shortwave infrared spectral band in MODIS. This band is not available in Sentinel-2 at 10 or 20 m of spatial resolution; $Slope = \frac{(\rho_{nir}-\rho_{swir2})}{(\lambda_{nir}(855.121\ nm)-\lambda_{swir2}(1627.713\ nm))}$; * NDWI was calculated with $\rho_{swir2}$ (band 11) in Sentinel-2 obtaining NDWI 2.

## 2.6. Measurement of LAI

A LAI-2200 equipment (LI-COR Inc., Lincoln, NE, USA) was used to collect LAI measurements over the growing period.

In both cultivated plots, BY1 and BY2, six points were selected to obtain LAI data, coinciding with the spectro-radiometric measurement points (Figure 2). In diffuse light conditions (dawn, dusk or very cloudy days), three LAI measurements were taken at the base of the plants that cover each of the six points.

These data were later processed through the FV2200 software (1.2.1. version, LI-COR Biosciences), in order to obtain a specific LAI value for each point and date. The average of the six points was calculated to obtain an integrated plot value.

## 2.7. Measurement of Soil Temperature and Soil Moisture Content

RT-1 and 10HS sensors (Decagon Services Inc., WA, USA) were used to collect soil temperature (°C) and soil moisture ($m^3\ m^{-3}$) content at an hourly interval at 10 cm depth in the three plots and registered in Em5b Analog Data Loggers (Decagon Services Inc., WA, USA). From the total data recorded, those hours corresponding to the soil respiration and spectro-radiometric measurements were selected.

## 2.8. Barley Biomass Determination

Barley biomass produced in each plot in both years was calculated by weighting the harvested plants within a surface equivalent to the field of view of the spectroradiometer (1 m height). This area corresponded to a circle of 22 cm radius which is equivalent to a surface of 0.15 $m^2$. Since a total of six angles have been measured, as indicated in Section 2.5, the surface sampled for each plot was 0.91 $m^2$.Plant cutting was done manually at a height of about 15 cm from the ground, simulating the conditions of a mechanical harvest (shoot biomass). The result was finally expressed in kg $ha^{-1}$.

Carbon fixation was estimated from the average value of biomass of the two plots of barley (BY1 and BY2). C percentage in barley of the biomass was estimated as Carvajal et al. [65]. These authors established a 43% of C in total shoot biomass and 28% C in total root biomass. Shoot to root biomass ratio was established according to Bolinder et al. [66] and Chirinda et al. [67] that considered a value of two.

## 2.9. Statistical Analysis

Linear regression models were used to assess the relationships between Rs, LAI and spectral indices. Measurements of both years and of all plots (BY1, BY2 and S plots) were taken into account to build the linear regression models. Additionally, soil temperature and moisture content were included to evaluate their significance in the models. The least square method was used to adjust the models and an analysis of variance (ANOVA) was applied to assess the significance of the model through the *F*-statistic. In addition, the *t*-statistic was used to assess the significance of model's coefficients. The proportion of variance explained by the regression models were measured and compared by the adjusted coefficient of determination ($R_{adj}{}^2$). The statistical analyses were performed through Statgraphics Centurion XVI (StatPoint Technologies Inc., Warrenton, VA, USA) and SAS software (SAS 9.4 Software, SAS Institute Inc., Cary, NC, USA).

## 3. Results

For representing the evolution of the variables during the growing period of barley, average values of the three plots were calculated. Two time series for each variable each year were obtained: one representing the average between cultivated plots (BY1 and BY2) and other for the bare soil (S). Table 3 shows the name of the time series of each variable.

**Table 3.** Names of the variables used in the plots.

| Variable | Cultivated Plots | Bare Soil |
| --- | --- | --- |
| Soil Respiration | Rs_BYyear | Rs_Syear |
| Index | Index_BYyear | Index_Syear |
| Leaf area index | LAI_BYyear | |
| Soil moisture | H10_BYyear | H10_Syear |
| Soil temperature | T10_BYyear | T10_Syear |

### 3.1. Dynamics of Rs and LAI

LAI and Rs dynamics are shown for both years in Figure 3a,b. LAI values were higher in 2012 than in 2013 with 3.48 $m^2 m^{-2}$ and 1.9 $m^2 m^{-2}$ maxima values occurring at the beginning of May (DOY 130) and June (DOY 157), respectively. The minimum occurred at the beginning of the growing season (March). LAI dynamics showed similarities between both growing seasons. Values increased sharply from sowing (S) to booting (B) during vegetative stages. After these vegetative stages, LAI values started to decrease during the reproductive stages until the soft dough development stage (D) or a bit later. Then, LAI values remained relatively constant during the rest of the stages of grain filling until maturity and harvest (H).

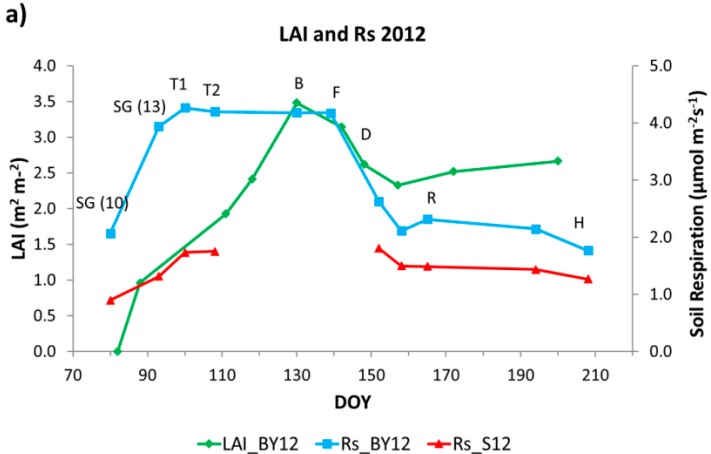

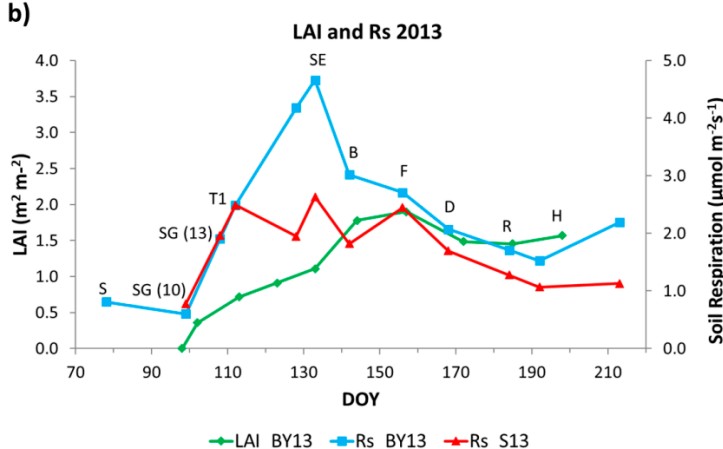

**Figure 3.** Leaf area index (LAI) and soil respiration (Rs) average values measuredin the BY1 and BY2 barley plotsand in the bare soil plot (S) in 2012 (**a**) and in 2013 (**b**) in Madrid, Spain.

Although LAI was higher in 2012 than in 2013, Rs in the cultivated plots (BY) had similar maximum values, around 4.5 µmol m$^{-2}$ s$^{-1}$ in both years. In 2012, the maximum value occurred at the beginning of April (DOY 100) while in 2013 occurred at mid-May (DOY 133). Rs dynamics in barley showed differences between years. In both years, Rs values increase during the first vegetative stages. Then, in 2012, Rs reached its maximum during tillering (T), one month earlier than maximum LAI (DOY 130) that occurred during stem elongation (SE) and booting (B). After that, values decreased slightly until flowering (F) and remained stable during approximately one month (April-May, DOY 100–139). In 2013, on the other hand, Rs values increased during tillering (T) until stem elongation (SE) when Rs presented a clear maximum (DOY 133), also around one month earlier than maximum LAI (DOY 157). After flowering (F) in 2012 and stem elongation (SE) in 2013, values decreased clearly during the first reproductive stages in 2012, and during booting (B) and also the first reproductive stages in 2013. In both years, minima values occurred at the beginning and the end of the growing season.

Rs in bare soil (i.e., Rh) showed also different dynamics between years. In 2012, Rs increased slightly during March and April. Then, there was a period without measurements for technical problems. After that, values remained constant during June and July. In 2013, Rs increased clearly during April. Then, Rs began to have an irregular pattern increasing and decreasing during April and May. Rs had its maximum value, 2.63 µmol m$^{-2}$ s$^{-1}$, in mid-May. After that, Rs decreased during June and July and remained constant until August.

### 3.2. Dynamics of Spectral Indices

The evolution of spectral indices for MODIS and for Sentinel-2 in 2012 and 2013 is shown in Figures 4 and 5, respectively. The spectral indices followed the evolution of LAI in both years with similar dynamics but different values between them.

**a)**

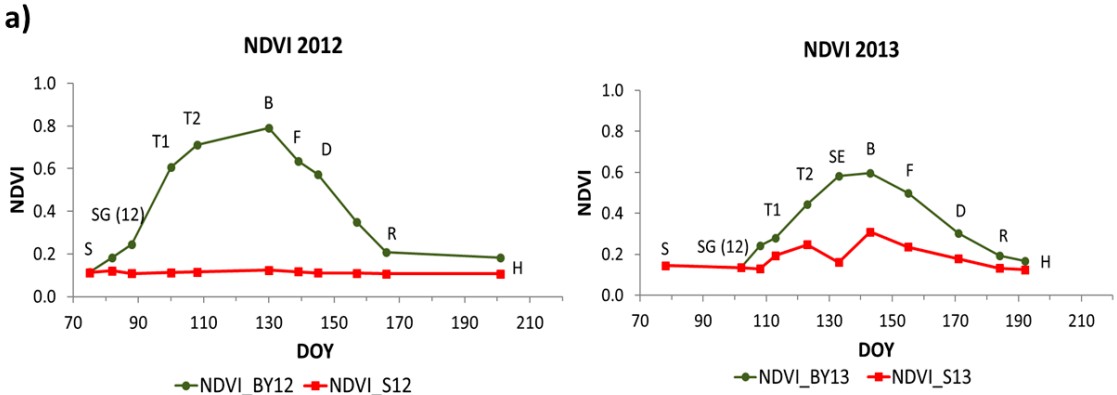

**b)**

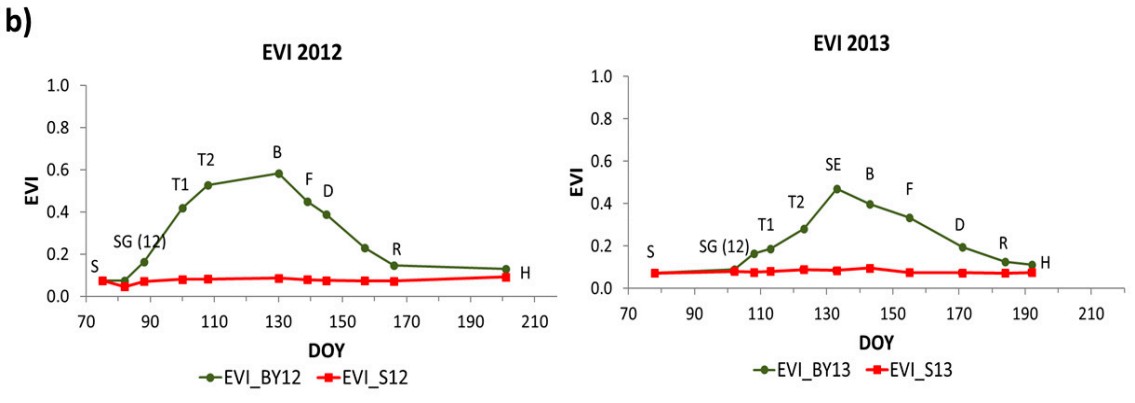

**Figure 4.** *Cont.*

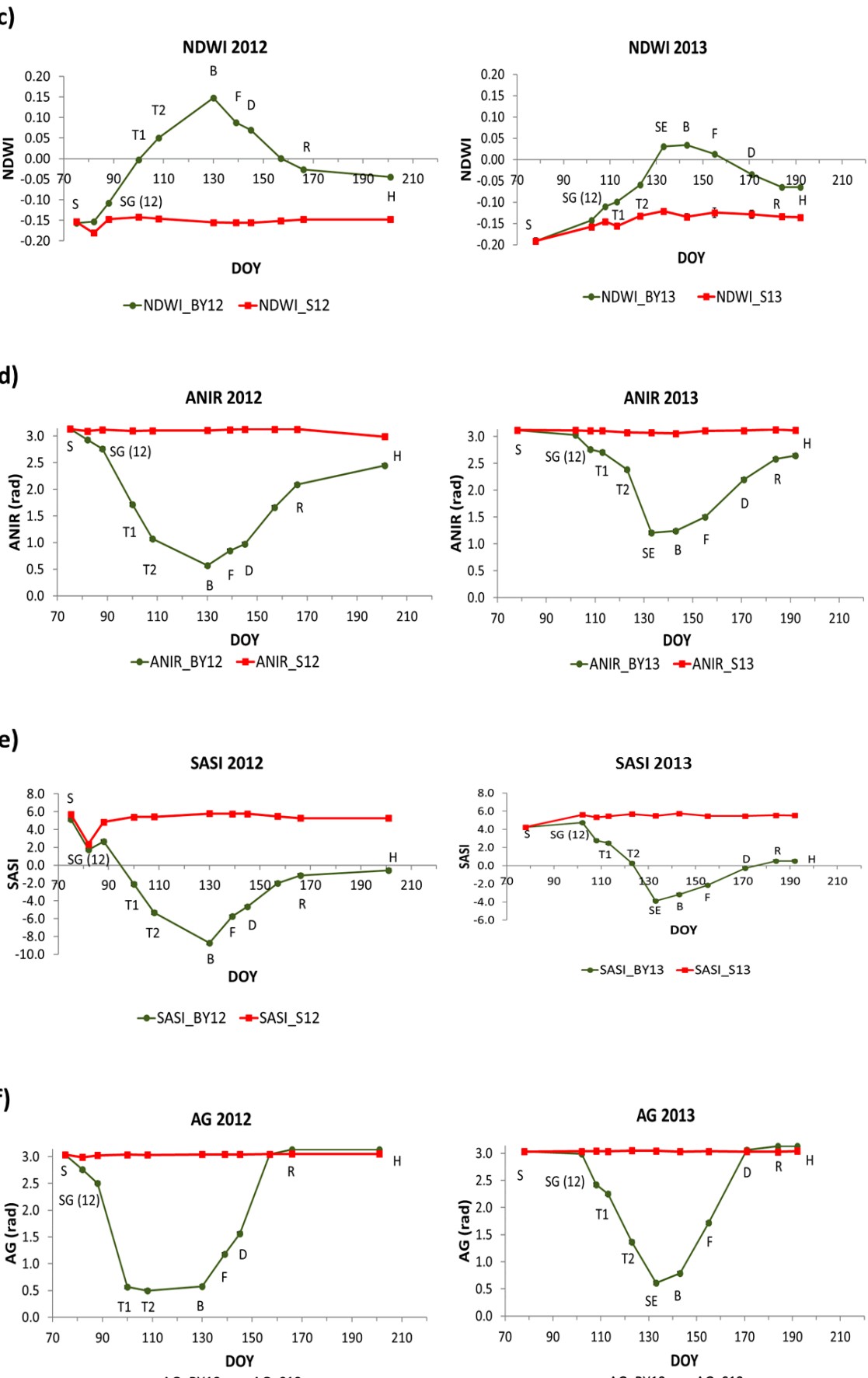

**Figure 4.** *Cont.*

**g)**

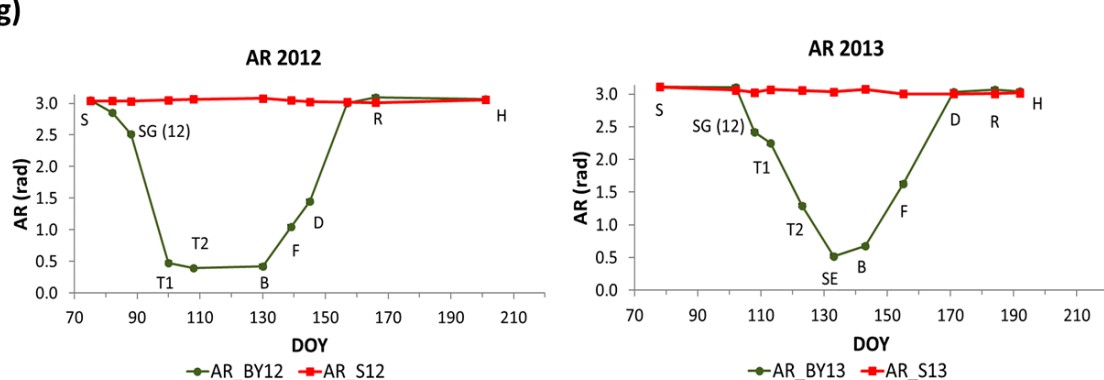

**Figure 4.** Spectral MODIS indices dynamics: (**a**) NDVI, (**b**) EVI, (**c**) NDWI, (**d**) ANIR, (**e**) SASI, (**f**) AG, (**g**) AR. Average values for the two barley plots (BY) and measured values for the bare soil plot (S) in 2012 and in 2013 in Madrid, Spain.

**a)**

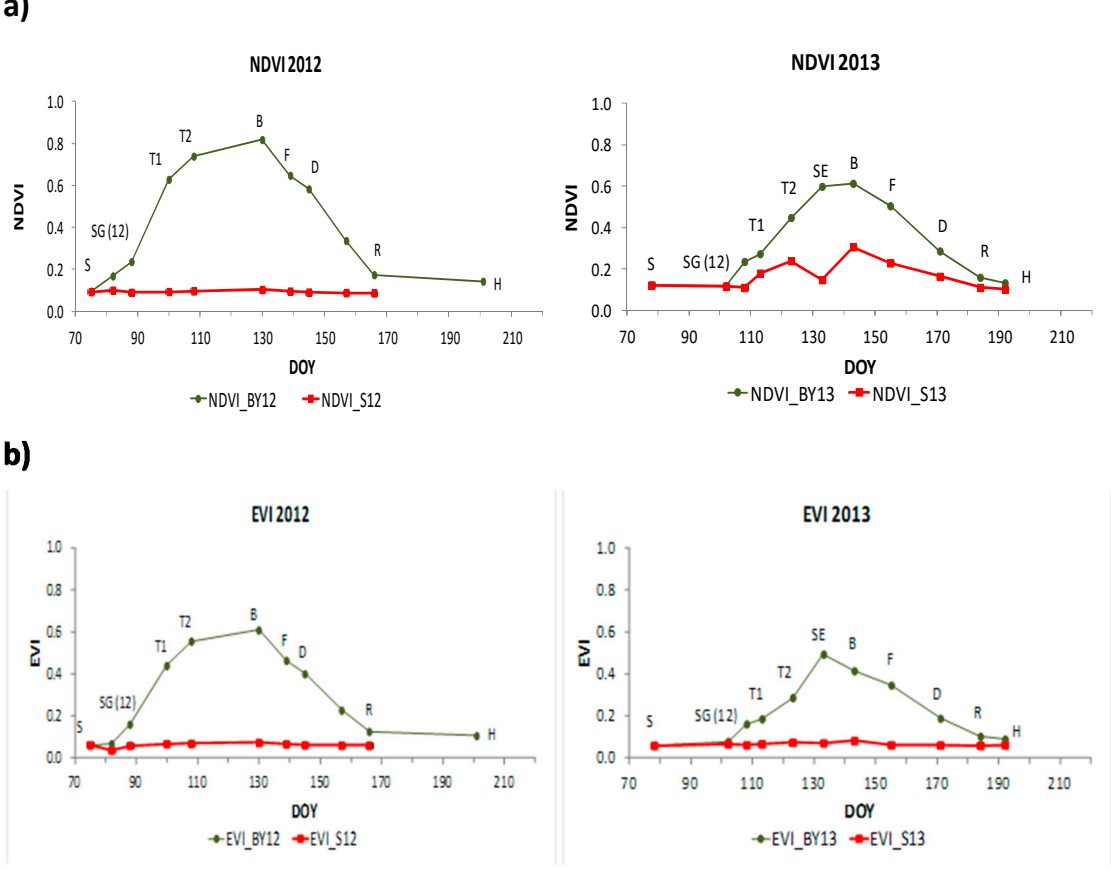

**Figure 5.** *Cont.*

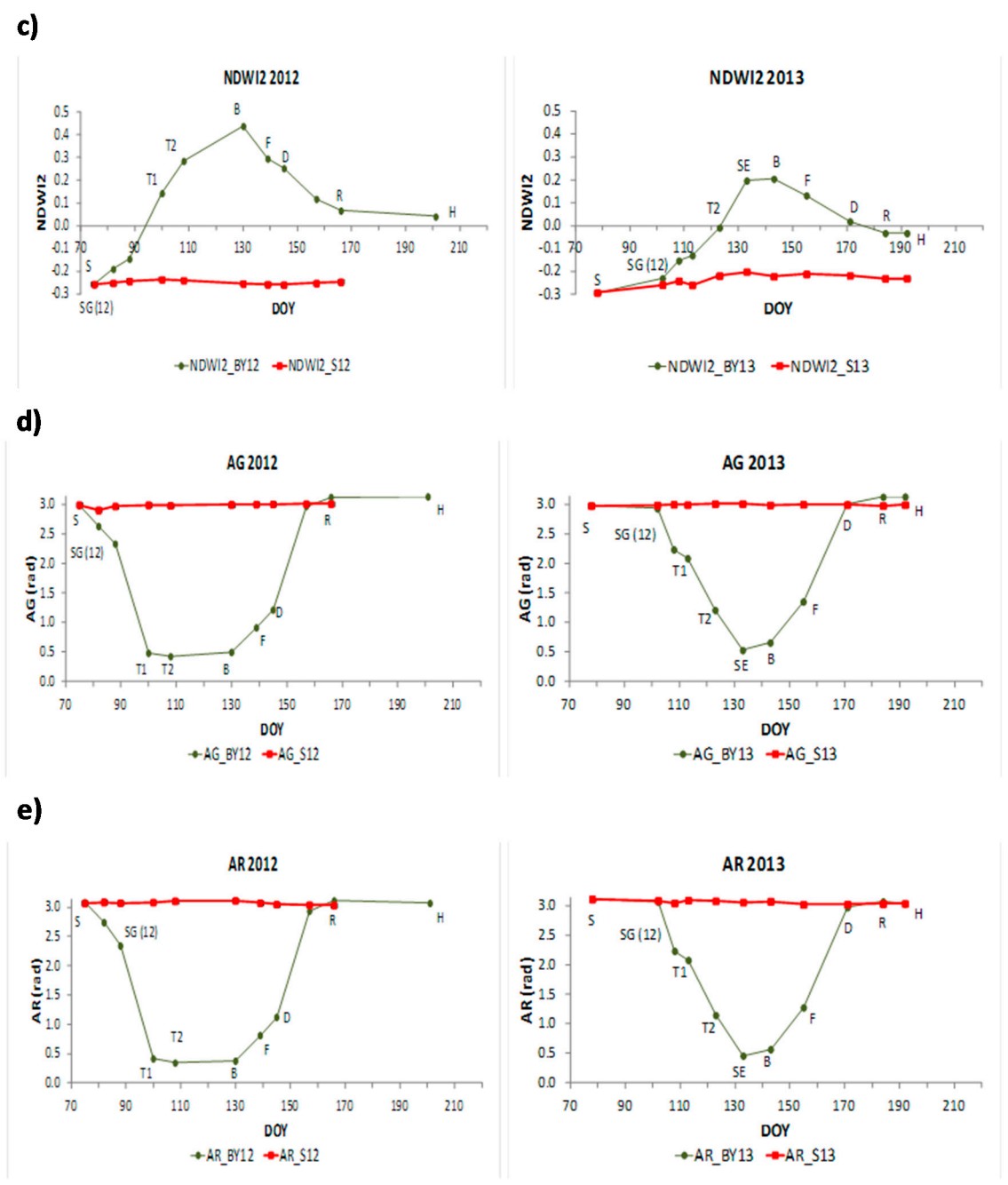

**Figure 5.** Spectral Sentinel-2 indices dynamics: (**a**) NDVI, (**b**) EVI, (**c**) NDWI2, (**d**) AG, (**e**) AR. Average values for the two barley plots (BY) and measured values for the bare soil plot (S) in 2012 and in 2013 in Madrid, Spain.

In both years, NDVI, EVI, NDWI and NDWI2 increased notably during the vegetative stages from emergence to booting (B) when they reached their maximum values, except EVI in 2013 which reached its maximum value before booting. Then, values started to decrease during reproductive stages until harvest reaching similar minimum values as in the first stages of barley. In NDWI, values at the end of the growing season were higher than values at the beginning of the growing season. NDVI, EVI, NDWI and NDWI2 in the bare soil plot remained regular all the period in both years with values around 0.1 in NDVI, 0.07 in EVI, −0.15 in NDWI and −0.3 in NDWI2.

The SSI ANIR, SASI, AG and AR showed similar dynamics as the previous indices, but they showed an inverse pattern. They presented maximum values at the first stages of barley and then

decreased reaching minimum values during booting (B) in 2012 and stem elongation (SE) in 2013. After that, they began to increase during grain filling stages until harvest reaching similar values as at the beginning of the period.

### 3.3. Dynamics of Soil Temperature and Soil Moisture Content

The dynamics of soil temperature (T10) and soil moisture (H10) at 10 cm depth for both years are shown in Figure 6a,b.

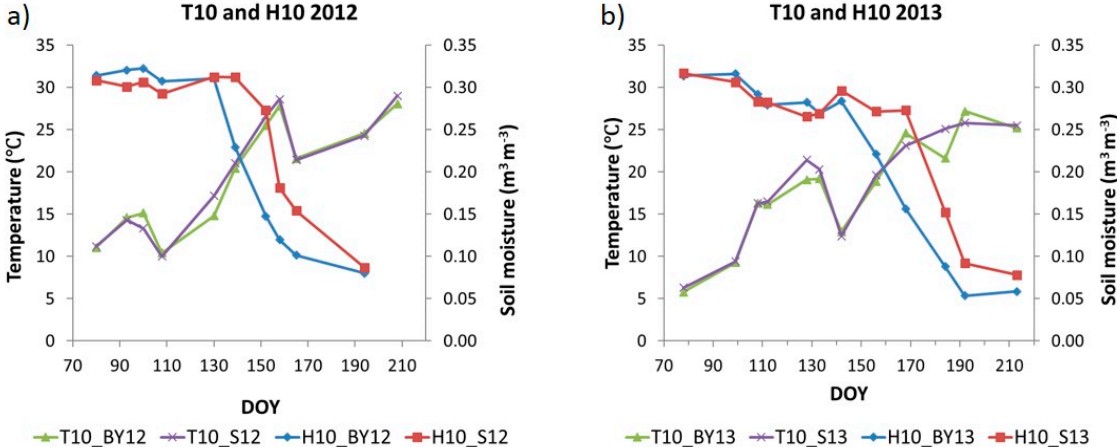

**Figure 6.** Soil moisture (H10) and Soil temperature (T10) dynamics at 10 cm depth. Average values for the two barley plots (BY) and measured values for the bare soil plot (S) in 2012 (**a**) and in 2013 (**b**) in Madrid, Spain.

The dynamics of H10was similar in 2012 and 2013 with higher values at early spring and had a clear decrease since mid-May until July. However, there were some differences between the two years. In 2012, soil moisture in cultivated plots had a maximum at the beginning of April (DOY 100) and a slight decrease after this date during the rest of April, although values were high and relatively constant, around 0.300 m$^3$ m$^{-3}$, during spring until the beginning of May (DOY 130). After this date, values showed a sharp decrease during May and June reaching values around 0.100 m$^3$ m$^{-3}$. In 2013, soil moisture was high in March until the beginning of April having a maximum of 0.316 m$^3$ m$^{-3}$ (DOY 99). After that, values began to decrease in mid-April and decreased during the rest of April and May. Although this decline was small, it was stronger than in 2012. Thereafter, values were constant until the end of May (DOY 142) when soil moisture decreased strongly as in 2012 until mid-July (DOY 192) reaching a minimum of 0.051 m$^3$ m$^{-3}$. In both years, H10 in the bare soil plot (S) had a similar dynamic as the cultivated plots; however, when the soil moisture began to decrease sharply, values were higher than in the latter plots (BY).

T10 in cultivated plots increased during spring and continued increasing during summer in June and July when maximum values were reached for both years. In 2012, soil temperature increased during March and the beginning of April. Values ranged from 11.2 °C in March (DOY 80) to a maximum of 15.2 °C in April (DOY 100). Then, values decreased during the rest of April reaching the minimum value of the growing period of 10 °C (DOY 108). After that, temperatures increased strongly during May and June reaching a maximum of 28.6 °C at the end of June (DOY 158). In 2013, T10 increased steadily during March, April and the beginning of May. Values ranged from the minimum value of the time series of 6.3 °C in March (DOY 78) to a maximum value of 19.2 °C at the beginning of May (DOY 133). After that, T10 decreased strongly at the end of May (DOY 142) reaching a local minimum of 12.4 °C but then temperature began to increase until mid-July reaching a maximum of 27.2 °C (DOY 192). In the bare soil plot (S) the dynamic was similar in both years having slightly higher values than the cultivated plots when temperature was increasing during spring.

### 3.4. Statistical Relationships between LAI and Spectral Indices

The coefficients of determination ($R_{adj}^2$) of a linear regression model between LAI and spectral indices from MODIS and from Sentinel-2 for the vegetative and the reproductive stages are shown in Tables 4 and 5, respectively. In MODIS, all linear regression models were significant according to the *F*-statistic and all the variables were significant in the models according to the *t*-statistic. The proportion of LAI's variance explained by the indices was higher during the vegetative stages, with $R_{adj}^2$ values ranging from 0.64 to 0.89, than in the reproductive period, with $R_{adj}^2$ values ranging 0.19 to 0.61. During vegetative stages NDVI, NDWI, EVI, ANIR and SASI explained a higher proportion of LAI's variance than AG and AR. During reproductive stages, SASI, NDWI and ANIR showed the highest values among all of them.

**Table 4.** Coefficients of determination ($R_{adj}^2$) of a linear regression model for both years between LAI and MODIS spectral indices (LAI = a + b ∗ Index) for the vegetative (*n* = 22) and the reproductive stages (*n* = 18) in a rainfed barley (*Hordeum vulgare* L.) in Madrid, Spain.

| MODIS Indices | AG | AR | ANIR | SASI | NDVI | NDWI | EVI |
|---|---|---|---|---|---|---|---|
| **Vegetative stages** | | | | | | | |
| LAI = a + b ∗ index | 0.64 | 0.66 | 0.80 | 0.84 | 0.85 | 0.89 | 0.83 |
| **Reproductive stages** | | | | | | | |
| LAI = a + b ∗ index | 0.19 | 0.19 | 0.45 | 0.61 | 0.22 | 0.53 | 0.27 |

Significance of model's coefficients was evaluated by the *t*-statistic ($p < 0.05$).

**Table 5.** Coefficients of determination ($R_{adj}^2$) of a linear regression model for both years between LAI and Sentinel-2 spectral indices (LAI = a + b ∗ Index) for the vegetative (*n* = 22) and the reproductive stages (*n* = 18) in a rainfed barley (*Hordeum vulgare* L.) in Madrid, Spain.

| Sentinel-2 Indices | AG | AR | NDVI | EVI | NDWI 2 |
|---|---|---|---|---|---|
| **Vegetative stages** | | | | | |
| LAI = a + b ∗ index | 0.64 | 0.65 | 0.84 | 0.82 * | 0.87 |
| **Reproductive stages** | | | | | |
| LAI = a + b ∗ index | 0.15 [I] | 0.16 [I] | 0.20 | 0.24 | 0.63 |

Significance of model's coefficients was evaluated by the *t*-statistic ($p < 0.05$). *: Intercept not significant. I: Index not significant.

In Sentinel-2, all linear regression models were significant according to the *F*-statistic, however, the intercept was not significant in the case of EVI during vegetative stages and AG and AR were not significant during reproductive stages according to the *t*-statistic. Similar results as those in MODIS were obtained. During vegetative stages NDVI, EVI and NDWI2 explained a higher proportion of LAI's variance than AG and AR. During reproductive stages, NDWI2 showed the highest value.

### 3.5. Statistical Relationships between Rs, LAI and Spectral Indices

The coefficients of determination ($R_{adj}^2$) of simple linear regression models between Rs and LAI, and between Rs and spectral indices, and of multiple linear regression models when soil temperature and soil moisture were included as independent variables are shown in Table 6 for MODIS and Table 7 for Sentinel-2 for the vegetative and the reproductive stages.

**Table 6.** Coefficients of determination ($R_{adj}^2$) of simple and multiple linear regression models for Rs, as a function of LAI, MODIS spectral indices, soil temperature (T10) and soil moisture (H10) for both years for the vegetative (*n* = 22) and reproductive stages (*n* = 18) in a rainfed barley crop (*Hordeum vulgare* L.) in Madrid, Spain.

| LAI and MODIS Indices | LAI | AG | AR | ANIR | SASI | NDVI | NDWI | EVI |
|---|---|---|---|---|---|---|---|---|
| **Vegetative stages** | | | | | | | | |
| Rs = a + b * Index | 0.42 | 0.56 | 0.57 | 0.43 | 0.46 | 0.52 | 0.44 | 0.53 |
| Rs = a + b * Index + c * T10 | 0.64 * | 0.64 | 0.64 | 0.61 | 0.64 * | 0.66 * | 0.59 * | 0.66 * |
| Rs = a + b * Index + c * H10 | - | - | - | - | - | - | - | - |
| Rs = a + b * Index + c * T10 + d * H10 | - | 0.71 * | 0.71 * | 0.69 * | 0.70 * | 0.71 | - | 0.73 * |
| **Reproductive stages** | | | | | | | | |
| Rs = a + b * Index | 0.62 * | 0.55 | 0.55 | 0.71 | 0.78 | 0.58 | 0.75 | 0.64 |
| Rs = a + b * Index + c * T10 | 0.72 | - | - | - | - | - | - | - |
| Rs = a + b * Index + c * H10 | 0.85 * | - | - | - | - | - | - | - |
| Rs = a + b * Index + c * T10 + d * H10 | - | - | - | - | - | - | - | - |

Significance of model's coefficients was evaluated by the *t*-statistic ($p < 0.05$). *: Intercept not significant. -: model or variables not significant.

**Table 7.** Coefficients of determination ($R_{adj}^2$) of simple and multiple linear regression models for Rs, as a function of LAI, Sentinel-2 spectral indices, soil temperature (T10) and soil moisture (H10) for both years for the vegetative (*n* = 22) and reproductive stages (*n* = 18) in a rainfed barley crop (*Hordeum vulgare* L.) in Madrid, Spain.

| LAI and Sentinel-2 Indices | LAI | AG | AR | NDVI | EVI | NDWI 2 |
|---|---|---|---|---|---|---|
| **Vegetative stages** | | | | | | |
| Rs = a + b * Index | 0.42 | 0.57 | 0.58 | 0.52 | 0.54 | 0.47 |
| Rs = a + b * Index + c * T10 | 0.64 * | 0.63 | 0.64 | 0.66 * | 0.66 * | 0.64 * |
| Rs = a + b * Index + c * H10 | - | - | - | - | - | - |
| Rs = a + b * Index + c * T10 + d * H10 | - | 0.70 * | 0.71 * | 0.71 | 0.73 | 0.70 * |
| **Reproductive stages** | | | | | | |
| Rs = a + b * Index | 0.62 * | 0.50 | 0.50 | 0.56 | 0.61 | 0.78 |
| Rs = a + b * Index + c * T10 | 0.72 | - | - | - | - | - |
| Rs = a + b * Index + c * H10 | 0.85 * | - | - | - | - | - |
| Rs = a + b * Index + c * T10 + d * H10 | - | - | - | - | - | - |

Significance of model's coefficients was evaluated by the *t*-statistic ($p < 0.05$). *: Intercept not significant. -: model or variables not significant.

In MODIS and in Sentinel-2, when only the index was included in the model, coefficients of determination were moderately high, around or above 0.5, with $R_{adj}^2$ similar or higher in the reproductive than in the vegetative period. In addition, all models were significant according to the *F*-statistic. Variables were significant in all cases throughout all the growing period. Along the vegetative period, the spectral indices explained a higher rate of Rs variance than LAI, while during

the reproductive stages, LAI showed a $R_{adj}^2$ higher than the spectral indices, except with ANIR, SASI, NDWI and EVI in the case of MODIS and except NDWI2 in the case of Sentinel-2.

In both cases, when T10was included in the models, coefficients of determination increased in the vegetative stages for all indices while in the reproductive stages they increased only for LAI. In general, during the vegetative stages, variables were significant while during the reproductive stages T10 was not significant in any of the spectral indices and the *F*-statistic values decreased when T10 was included.

In both cases, H10 was not significant according to the *t*-statistic during the vegetative stages, in addition, the $R_{adj}^2$ and *F*-values decreased in all cases when the models took into account soil moisture. In the reproductive stages, $R_{adj}^2$ values were higher than in the vegetative stages, but $R_{adj}^2$ and *F*-statistic values were in general lower than when H10 was not included. In addition, soil moisture was not significant in all spectral indices. However, it has to be remarked that LAI plus H10 explained the highest proportion of Rs variance (0.85).

In MODIS and in Sentinel-2, when the models included together both abiotic factors, $R_{adj}^2$ values increased in the vegetative stages and *F*-values remained significant in all cases. It has to be remarked that H10 was not significant when it was included alone with the index, however, when it was included together with T10 it was significant for all indices except LAI andin MODIS NDWI. In the reproductive stages, $R_{adj}^2$ values were lower than when only the index was included in the model and in general lower than in the vegetative stages, *F*-values also decreased. In addition, neither T10 nor H10 were significant in all indices except in the LAI in which soil moisture was significant.

Table 8 shows the coefficients of determination ($R_{adj}^2$) of the simple linear regression models between Rs and soil temperature (T10), between Rs and soil moisture (H10), and of multiple linear regression models when both variables were considered as independent variables, for both stages in the cultivated plots. On one hand, T10 showed low $R_{adj}^2$ values during the vegetative stages and it did not explain Rs variance during the reproductive stages. In both cases, T10 was significant according to the *t*-statistic. On the other hand, H10 showed moderate values in the reproductive stages, and null values in the vegetative stages. When both variables were included together in the models $R_{adj}^2$ values did not increase.

**Table 8.** Coefficients of determination ($R_{adj}^2$) of the simple and multiple linear regression models between soil respiration(Rs) and soil temperature (T10) and soil moisture (H10) at 10 cm depth for both years for the vegetative (*n* = 22) and reproductive stages (*n* = 18) and $R_{adj}^2$ of the simple and multiple linear regression models between Rs in bare soil (Rh) and T10 and H10 for 2012 and 2013, for the vegetative (*n* = 11) and reproductive stages (*n* = 9) separately in a rainfed barley (*Hordeum vulgare* L.) in Madrid, Spain.

| | T10 | H10 | T10+H10 |
|---|---|---|---|
| **Vegetative stages** | | | |
| Rs | 0.22 * | 0 | 0.23 *,H |
| **Reproductive stages** | | | |
| Rs | 0.02 * | 0.38 | 0.35 *,T |
| **Bare soil (Rh)** | | | |
| Rh (vegetative stages) | 0.41 * | 0.02 *,H | 0.39 *,H |
| Rh (reproductive stages) | 0.25 T | 0.52 * | 0.57 *,T |

Significance of model's coefficients was evaluated by the *t*-statistic ($p < 0.05$). *: Intercept not significant. T: T10 not significant. H: H10 not significant.

Table 8 also shows the results in the bare soil plots, including $R_{adj}^2$ values of the simple linear regression models between Rs, i.e., heterotrophic respiration (Rh), and soil temperature (T10), between Rs and soil moisture (H10), and of multiple linear regression models when T10 and H10 were considered along vegetative and reproductive stages. If the growing period was divided according to the vegetative and reproductive stages in the crop the $R_{adj}^2$ increased. In the vegetative stages, T10 was significant and presented a moderate $R_{adj}^2$ and H10 was not significant either alone or together with

T10. Meanwhile, in the reproductive stages soil moisture was significant and presented a moderate $R_{adj}^2$, and although T10 showed a low $R_{adj}^2$ alone and increased the $R_{adj}^2$ together with H10, it was not significant in the models.

*3.6. Relationship between Soil $CO_2$ Efflux and Biomass*

Total $CO_2$ emissions for the growing cycle were 1505.16 g $CO_2$ m$^{-2}$ in 2012 and 1009.28 g $CO_2$ m$^{-2}$ in 2013, corresponding to410.91g of C m$^{-2}$ and 275.53 g of C m$^{-2}$ respectively. The aboveground biomass at harvest was 8310.1 kg ha$^{-1}$ in 2012 and 4700 kg ha$^{-1}$ in 2013. Thus, the final value of C fixation was 473.67 g C m$^{-2}$ in 2012 and 267.9 g C m$^{-2}$ in 2013 resulting in estimated $C_{emitted}/C_{fixed}$ ratios of 0.87 in 2012 and 1.03 in 2013.

## 4. Discussion

During vegetative stages, Rs increased as a consequence of barley growth including the development of the root system (Figure 3). During reproductive stages, Rs decreased due to senescence of barley, which coincides with the decrease in soil moisture availability (Figure 6). This is in accordance with the expected evolution of both components of Rs: (1) Ra increases due to crop growth and then decreases due to senescence and (2) Rh increases together with soil temperature in spring and decreases together with soil moisture in summer. Jongen et al. [68] found that ecosystem respiration and Gross Primary Production decreased with the canopy senescence in Mediterranean grasslands. Similar patterns were found in irrigated crops [38,46]. Other authors attribute this behavior to a decrease in photosynthesis rate [69] and a reduction of C translocation to the root system during the grain filling processes [70,71].

Significant differences in LAI, Rs, and indices between years were found which is consistent with the high inter-annual variability in crop progression, typical of Mediterranean regions [72,73]. Spring barley development during vegetative stages required optimal temperature and water availability [74]. The adequate conditions of temperature and moisture at the beginning of the growing season in 2012, permitted a fast increase in Rs values, due to a higher activity of metabolism and a faster development [74,75]. On the other hand, anomalous low temperatures in 2013 March-April resulted in a delay of crop development at the beginning of the growing period (Figure 3b), which resulted in lower Rs, indices, LAI and biomass values throughout the crop cycle. Optimal moisture and temperature conditions of May 2013 gave rise to high Rs values later in the growing season. This is probably more associated to heterotrophic processes, so that soil heterotrophic communities could take advantage of such environmental conditions to a greater extent than the crop. These patterns showed the key role of the moisture-temperature timing in crop development [76,77] and, specifically, its effect on soil respiration [8,78].

During vegetative stages significant crop structural changes resulted in a high proportion of LAI variance (Tables 4 and 5) explained by NDVI, EVI, NDWI, NDWI2, ANIR and SASI i.e., those indices with influence of the NIR band. However, AG and AR, centered in the VIS spectral region, which is more related to functional traits [79], showed more moderate $R_{adj}^2$ values. During reproductive stages, with smaller LAI variation, only NDWI, NDWI2, ANIR and SASI (i.e., those with higher influence of the NIR-SWIR spectral region) maintained moderate values being higher in NDWI and SASI from MODIS and NDWI2 from Sentinel-2, which are the only indices that do not include the red band. This is in accordance with other researches which showed the relevance of the NIR-SWIR spectral region for assessing structural changes during crop development [80,81].

The proportion of Rs variance explained by LAI indicated that there must be other sources of variability that have a significant influence on soil $CO_2$ efflux. This fact explained the increase in $R_{adj}^2$ values when including soil temperature and moisture in the models of the vegetative and reproductive stages respectively, confirming the limiting role of temperature at the beginning of the growing period and soil moisture at the end [74]. In addition, high LAI $R_{adj}^2$ values during reproductive stages suggest that significant structural changes due to flowering and senescence are linked to Rs.

In both stages, some spectral indices showed significantly higher $R_{adj}^2$ values than LAI alone indicating that they contained additional information (probably functional). During the vegetative stages, NDWI, ANIR and SASI from MODIS and NDWI2 from Sentinel-2, showed coefficients of determination similar to those of the LAI, corroborating the strong relationship between LAI and these indices [52,82]. Indices centered in the VIS spectral region (i.e., AG, ARfrom both sensors), highly related to functional traits such as pigment content and photosynthesis [54,83], showed slightly higher values than LAI and the rest of the indices. This agrees with the evolution of these indices showing similar dynamics as Rs with distinct differences between years (Figures 4 and 5). By including the temperature in the AG and AR models the $R_{adj}^2$ improvement was smaller than in the case of LAI (Tables 6 and 7, second row). This fact suggests that these indices were already accounting for temperature effects in vegetation photosynthetic activity and emphasizes the importance of this variable on Rs variability at the beginning of spring. The results shown in the bare soil plots (Table 8) corroborated that temperature had a strong influence also in heterotrophic respiration. On the other hand, moisture did not seem to play an important role in Rs during the vegetative stages (Tables 6 and 7, third row), indicating that moisture is not a limiting factor either for the crop or for the soil microorganisms, as shown in Table 8.

The senescence process and drought stress resulted in alterations in leaf cell structure and composition [84,85] and a decrease in leaf moisture that has been well captured by the NDWI [64,86] and SASI. In our experiment we found high NDWI, SASI and NDWI2 $R_{adj}^2$ values with both LAI (Tables 4 and 5) and Rs, indicating that the capability of these indices to capture variability in vegetation structure and moisture could be the basis in using them for assessing Rs. The importance of moisture when only heterotrophic respiration is considered (Table 8, $R_{adj}^2 = 0.52$) decreases when the crop is taken into account ($R_{adj}^2 = 0.38$) evidencing the importance of crop senescence in soil $CO_2$ efflux dynamics. During this period ANIR showed also a high $R_{adj}^2$ value indicating that the structural information based on ANIR variability could help Rs assessment.

The difference between years in carbon emitted versus carbon fixed could be associated to abiotic factors dynamics; better conditions at the beginning of the growing cycle in 2012 resulted in a clear advantage for the plant to produce higher biomass. The inter-annual variability in the carbon flux balances has been shown also by other studies especially in Mediterranean climate regions [87–89]. In this aspect, taking into account photorespiration and root biomass values would be a significant contribution to more accurate estimations.

Spectral indices from MODIS and Sentinel-2 bands have shown to partially represent change in soil respiration, a highly dynamic process through the growing period, based on their link with crop physiology. While they did not provide direct information on soil microbial processes, including soil moisture and temperature, which are more easily estimated than soil respiration, improved the results.

It is recognized that measurements taken in this study, mostly throughout time, were subjected to auto-correlation at some degree. Effectively, and for the experimental layout adopted, results do not reflect the ability of spectral indices to assess Rs directly, but its change through time. Due to limitations to accurately assess soil respiration at large scales and long time periods, this study should be regarded as preliminary and providing useful insights on the combination of remote sensing data with soil temperature and moisture to potentially assess soil respiration variability across crops and crop stages. Future studies should gather more information across a range of environments (e.g., different soil types) and through longer time periods to develop robust and representative statistical models for Rs assessment.

## 5. Conclusions

This study showed that spectral indices improve the information provided by LAI to assess soil respiration in a rainfed barley crop. Soil moisture and temperature were relevant variables to complement spectral indices and/or LAI to estimate Rs in specific periods. During vegetative stages, indices together with soil temperature showed similar results than LAI with soil temperature, while including soil moisture with LAI improved Rs estimations during senescence.

Indices including the near infrared spectral region (ANIR, NDVI, EVI, NDWI, NDWI2 and SASI) were the most related to LAI during vegetative stages. Among them, moisture indices, i.e., SASI and NDWI from MODIS and NDWI2 from Sentinel-2, showed the best relationship with LAI during reproductive stages, indicating the impact of declining moisture during senescence on plant structure.

AG and AR indices, centered in the visible spectral region, showed the best relationship with Rs during vegetative stages characterized by active photosynthesis. On the other hand, the NIR-SWIR spectral indices (especially SASI from MODIS and NDWI2 from Sentinel-2) showed the highest relationships with Rs during reproductive stages showing the impact of plant senescence. Furthermore, SASI (angle centered at SWIR1) showed better results than NDWI, probably because it provides information farther in the shortwave spectral region highly related to water content.

This study showed the potential capability of spectral information to assess soil respiration linked to crop physiology. Our results based on measurements acquired through field spectroscopy are a first step to use these models across several spatial and temporal scales to monitor crop C balances under high inter-annual climate variability that result in distinct $C_{emitted}/C_{fixed}$ ratios between years. Our study emphasizes the usefulness of a frequent monitoring system in Mediterranean agricultural environments where equilibrium between fixation and emission is particularly fragile, especially in the context of climate change.

**Author Contributions:** Conceptualization, V.C., M.R.-R., M.H., T.S., R.I., A.P.-O.; methodology, V.C., M.R.-R., L.R., M.H., T.S., R.I., V.S.-G.; software, T.S., R.I., J.L., A.P.-O.; formal analysis, V.C., M.R.-R., L.R., M.H., J.L.; investigation, V.C., M.R.-R., L.R., A.P.-O.; resources, T.S., R.I., A.P.-O.; writing—original draft preparation, V.C., M.R.-R., L.R.; writing—review and editing, V.C., M.R.-R., L.R., J.L., V.S.-G., A.P.-O.; supervision, A.P.-O.; project administration, T.S., R.I., V.S.-G., A.P.-O.; funding acquisition, T.S., R.I., A.P.-O. All authors have read and agreed to the published version of the manuscript.

**Funding:** This research was funded by the Ministerio de Ciencia e Innovación (Spain) through the projects AGL-2010-17505 and CGL-2009-07031. This work was also funded supported by the Ministerio de Ciencia, Innovación y Universidades (Spain) through the pre-doctoral scholarship of Laura Recuero (Becas de Formación Profesorado Universitario, BOE-A-2015-9456).

**Acknowledgments:** We would like to thank the College of Agricultural Engineering of the Universidad Politécnica de Madrid for the concession of their experimental fields to carry out this research and to thank all the staff, especially Román Zurita, for their work in planting and taking care of the crop during both years. We would also like to acknowledge David Manrique for his collaboration in the field work.

**Conflicts of Interest:** The authors declare no conflict of interest. The funders had no role in the design of the study; in the collection, analyses, or interpretation of data; in the writing of the manuscript, or in the decision to publish the results.

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
