# Peer review of "First Insights on Soil Respiration Prediction across the Growth Stages of Rainfed Barley Based on Simulated MODIS and Sentinel-2 Spectral Indices"

_remotesensing, doi:10.3390/rs12172724_

Round 1

Reviewer 1 Report

The study used in-situ remote sensing along with other surface measurements of LAI and CO2 flux to assess how addition of the remote sensing data helped in quantifying the exchange of CO2 over a barley canopy at various stages in its development.

The paper is well-written, well-organized, straightforward in its objectives, and clearly presents its results.  I read though it several times and really couldn't find much to comment on.  The objective and results are similar to other remote sensing work for quantifying gas exchanges, but the combination of crop, phenological coverage, climate regime, and specifics of the spectral data used warrant publication of these results.  I have some very minor suggestions, but otherwise I believe the article is ready for publication.

1/  Lines 204-206.  I am not sure what is meant by resampling to 10m and 20m resolution.  Does this mean that the various point samples collected with the ASD were spatial extrapolated to these scales?  It's noted that these match MODIS and Sentinel data, but again this is not entirely clear.  Please clarify what is meant here.

2. Some clarification on how the ASD data was used to simulate the spectral ranges of Sentinel and MODIS data would also be helpful.  References are included, but a brief description would not add much to the length of the articles and help the reader avoid a bit of "reference chasing."

Author Response

Dear Reviewer 1,

Thank you very much for your revision and suggestions. You recommend us mainly two minor revisions:

  1. Lines 204-206. I am not sure what is meant by resampling to 10m and 20m resolution.  Does this mean that the various point samples collected with the ASD were spatial extrapolated to these scales? 
  2. Some clarification on how the ASD data was used to simulate the spectral ranges of Sentinel and MODIS data would also be helpful.

We have modified our text in material and method section to clarify these issues in lines 204-212 with the following text:

"Then, a resampling method to MODIS and to Sentinel-2 bands with the function spectral resampling of the software ENVI 4.2 (Research Systems Inc., Boulder, CO, USA) was done. In the case of MODIS, 36 bands were obtained and only the first seven spectral bands (B1-red, B2-nir, B3-blue, B4-green, B5-swir1, B6-swir2 and B7-swir3) were selected. In the case of Sentinel-2, the resampling method was done to seven spectral bands which have different spatial resolution 10, 20 and 60 meters (B2-blue (10 m), B3-green (10 m), B4-red (10 m), B8-nir (10 m), B10-swir1 (60 m), B11-swir2 (20 m) and B12-swir3 (20 m)). Finally, spectral indices were obtained (Table 2) for each of the six points and an average of the six points was calculated obtaining one measurement for each date."

We hope that these changes would be sufficient to clarify the article,

Best regards

Reviewer 2 Report

This is a resubmitted version and as I have already mentioned in my previous comment the paper provides some first insights for the prediction of soil respiration based on multispectral spaceborne data. The topic is really interesting and up to date in the domain of Earth Observation. However, I strongly believe that the limitation of this study should discussed in more detail, e.g. few samples, restricted area and only one. In that regard, recommend to re-phrase the title in order to indicate that the current study provides some “first insights for soil respiration prediction based on simulated multispectral spaceborne data”. Nearly all comments have been dealt with in a satisfactory manner, considering my previous review round.

Author Response

Dear Reviewer 2,

Thank you very much for your comments and your cooperation. We think that you have mainly two considerations:

  1. I strongly believe that the limitation of this study should discussed in more detail, e.g. fewsamples, restricted area and only one.

Answer

This is an experiment in the field with the common limitation of these type of experiments. However, only statistically significant results have been reported and discussed.

Some limitations are explained in the discussion section in the last paragraph.

  1. Re-phrase the title in order to indicate that the current study provides some “first insights for soil respiration prediction based on simulated multispectral spaceborne data”. 

Answer

We propose this new title:

¨First insights for soil respiration prediction across the growth stages of rainfed barley based on simulated MODIS and Sentinel-2 spectral indexes¨

We hope you like the new title and the changes we have made in the discussion section.

Best Regards

Reviewer 3 Report

It can be accepted as it is

Author Response

Dear Reviewer 3,

Thank you very much for your revision and your help. We have made some changes in the paper to improve its quality following the suggestions of the other reviewers. 

Best regards

Reviewer 4 Report

This manuscript has minor changes only from its previous version. The principal idea is to allow future and historical estimates of soil respiration at scale from satellite remote sensing data.

The simulated satellite indices used basically measure ‘greenness’ (e.g. NDVI) or ground cover (e.g. SASI). None of them have any temproal change in the bare soil plot, effectively being zero as would be expected for vegetation indices.

The principal of the work, as I understand it, is that as soil respiration is likely to be linked to aboveground biomass to some extent, it may be estimated using one or more of these indices.

The inability of the indices to pick up any changes in bare soil respiration, despite those changes being larger than under a sward, demonstrate the inability of any of the indices to effectively estimate soil respiration per se. This point is not clear in the manuscript as it stands.

The authors demonstrate that LAI is able to be used to predict soil respiration to some extent (adj r2 = 0.42), though it should be remembered that n=2 and the variation in data is only temporal, so there will be a large amount of auto-correlation in this data set for all of the measured parameters. Effectively, the result is not estimating soil respiration, but the change in soil respiration with time. Again, this is not made clear in the manuscript as it stands.

When the indices are compared to LAI for their ability to predict soil respiration, there is a limited improvement, mainly in AG and AR. I guess this is may be because these indices are better estimating the photosynthetic capacity of the canopy than LAI is. On line 488 the authors suggest it is because these indices are somehow accounting for temperature. I very much doubt this, especially given they are using RGB.

Visually comparing the soil respiration data with the soil temperature and soil moisture data suggests that temperature is (unsurprisingly) a major driver of respiration. Including temperature in the regression model improves the estimate of respiration, with no real difference between LAI or any of the indices any more. Adding soil moisture in helps in the first part of the season, but not the latter.

In regards to application, what do we have? The suggestion that MODIS or Sentinel 2 indices could be used to estimate the change in soil respiration under a crop that is dependant on the biomass/photosynthetic capacity of that crop. This is already inherent in the indices themselves as they were developed to estimate biomass.

We know from a vast literature on soil respiration that it is a lot more complicated than estimating with aboveground biomass and the authors show this by adding in soil temperature and moisture into their model (that isn’t available with MODIS or Sentinel-2). Basically, my comment on the previous version still stands, this manuscript would be much more valuable if the analysis went further.

As an additional comment, it would be nice to see some plots of some of the regression models, even if supplementary data so the reader can better judge the effect of the two reps and the bare soil data for instance.

Author Response

Dear Reviewer 4,

Thank you for your constructive suggestions, your work and cooperation.

We think that this article provided first insights to assess Rs linked to crop phenology based on spectral indices. Although the study is a field work that showed limitations, we think that this research is a prior step for upscaling the Rs models using satellites in agricultural crops. We agree that spectral indices do not estimated total soil respiration, however based on their link with crop physiology, are good indicators of much of soil respiration process. At the present it is unfeasible to estimate accurately sol respiration at large scales and long time periods on a frequent basis; thus, this approach could be a first step to develop operative methods based on remote sensing data and abiotic factors such as soil moisture and temperature, which are variables more easily estimated at different scales than direct soil respiration measurements.

We think that we have provided new relationships between spectral indices, abiotic factors and soil respiration. In addition, we have assessed several different spectral indices, some of them not so common such as Spectral Shape Indices and we have gone further including Sentinel-2 indices as many reviewers demanded. Although we are conscious that further research is needed and our article only provided first insights in the topic, we think that this article is suitable for publication.

We have taken into account the majority of your suggestions and we would like to comment some of them in the following lines. We have tried to answer your comments pont by point as follows.

  1. …..the inability of any of the indices to effectively estimate soil respiration per se. This point is not clear in the manuscript as it stands.

Answer

This is true, while the indexes provide information on the crop physiological processes through the growing period, they do not provide it on soil microbial processes, so that including soil moisture and temperature can improve the estimations. At the present it is unfeasible to estimate accurately sol respiration at large scales and long time periods on a frequent basis; thus, this approach could be a first step to develop operative methods based on remote sensing data and abiotic factors such as soil moisture and temperature, which are variables more easily estimated at different scales than direct soil respiration measurements.

This issue is clarified in lines 512-519 in the discussion section and in lines 539-540 in the conclusion section.

  1. …..though it should be remembered that n=2 and the variation in data is only temporal, so there will be a large amount of auto-correlation in this data set for all of the measured parameters.

Answer

Although the number of plots is 2, the number of measurements for building the models for both years is 22 during vegetative stages and 18 during reproductive stages. However, we agree that it would be interesting to extend this research to bigger agricultural areas with more plots to represent better spatial variability and for a longer period of time to avoid amount of auto-correlation in the measured parameters.

This fact has been discussed in lines 519-521.

  1. Effectively, the result is not estimating soil respiration, but the change in soil respiration with time. Again, this is not made clear in the manuscript as it stands.

Answer

Measurements were done at a specific time of the day and location, so that the results estimate the soil respiration at these conditions which may not be completely representative of the real situation, which is the main limitation of most of the research based on field data.

However, the evolution in time provides useful insights to understand how remote sensing data could help to monitor soil respiration variability through the crop cycle and will be useful to compare between crop periods and crops.

These comments have been included in lines 516-519 in the discussion section.

  1. When the indices are compared to LAI for their ability to predict soil respiration, there is a limited improvement, mainly in AG and AR. I guess this is may be because these indices are better estimating the photosynthetic capacity of the canopy than LAI

Answer

We agree with the reviewer and we think it is consistent with the comment in line 491. We considered that since photosynthetic activity is highly mediated by temperature those indexes were already taking into account the effect of temperature on photosynthesis. To clarify this issue, we have changed “functional processes” by “photosynthetic activity”.

  1. The suggestion that MODIS or Sentinel 2 indices could be used to estimate the change in soil respiration under a crop that is dependant on the biomass/photosynthetic capacity of that crop. This is already inherent in the indices themselves as they were developed to estimate biomass.

Answer

We agree, so that the objective of this research was to assess quantitively this relationship based on field measurements to create knowledge for future applications. We think that this fact is already clarified in the objectives of the paper and that has been clarified in lines 515-519 in the discussion section.

We look forward to hearing from you soon, thanks again for your work. We think that the article have improved a lot with your comments.

Best regards

Reviewer 5 Report

Review report: remotesensing-869291

Manuscript Number: remotesensing-869291

Title: Vegetation indices from MODIS and Sentinel-2 spectral wavelengths predict soil respiration across the growth stages of rainfed barley

General remarks

This study provided methods, results, and analysis are potentially useful for science and society. Thus, I recommend this manuscript for acceptance for publication in the journal of Remote Sensing after Minor revision from the authors.

Detailed comments

Title

The title of the manuscript is fine and it cover overall content of the manuscript.

Abstract

 Line 24-26: “Rainfed agriculture occupies 80 % of the world’s agricultural surface which is expected to increase in the near future causing serious effects on carbon cycle dynamics in the context of climate change”. This statement have exact figure, and it not based on this study, thus it needed reference. Therefore, I suggest to rephrase this statement without exact figure.

There is lack of specific results with exact value of the study in abstract. Add some specific results with exact value of the study.

Keywords: Reduce keywords within five keywords.

  1. Introduction

The overall write-up of the introduction is fine.

  1. Materials and Methods

Line 120: The study was carried out in a rainfed barley field during two crop seasons (2012 and 2013). I am wondering why the authors submitted there manuscript after 7 years of data collection? Can you please explain briefly regards this?

Line 132-133: The area has a Mediterranean continental climate (Csa according to Köppen-Geiger classification). It needed citation/reference.

Line 133-134: During the barley growing cycle, average temperatures range from 8.0 0C (March) to  20.1 0C (June). It needed citation/reference, without data source and citation how the readers knows its reliability.

Figure 1: Please add global map in the left top side and highlight country with name. In this way, readers will clearly noticed the geographical area of the study site.  

  1. Results

The results of the study fine and written in a good way.

  1. Discussion

The overall analysis of the discussion is fine.

  1. Conclusions

The conclusions of the manuscript little lengthy. I suggest to present only concise conclusion within a paragraph.

Author Response

Dear Reviewer 5,

Thank you for your extensive review. We hope that our changes improve the quality of the paper with your suggestions. Your main suggestions are the following:

1. Abstract

 Line 24-26: “Rainfed agriculture occupies 80 % of the world’s agricultural surface which is expected to increase in the near future causing serious effects on carbon cycle dynamics in the context of climate change”. This statement have exact figure, and it not based on this study, thus it needed reference. Therefore, I suggest to rephrase this statement without exact figure.

There is lack of specific results with exact value of the study in abstract. Add some specific results with exact value of the study.

Answer

We have rephrased the statement in lines 24-26 and we have eliminated the number of 80% that would need a reference:

"Rainfed agriculture occupies the majority of the world’s agricultural surface and is expected to increase in the near future causing serious effects on carbon cycle dynamics in the context of climate change."

We think that we have already specify some results of our study, however we have changed some words in the abstract in lines 32,33 and 40.

2. Keywords: Reduce keywords within five keywords.

Answer

We have reduced from seven keywords to five.

3. Materials and Methods

Line 120: The study was carried out in a rainfed barley field during two crop seasons (2012 and 2013). I am wondering why the authors submitted there manuscript after 7 years of data collection? Can you please explain briefly regards this?

Answer

The field work ended at 2013. However, then we had to process all the data, built statistical models and wrote the paper. This was not easy because we did not have economical and human resources. Finally the paper was done because a phd student recollected the data and began to write it to made his thesis.

Line 132-133: The area has a Mediterranean continental climate (Csa according to Köppen-Geiger classification). It needed citation/reference.

Answer

We have added reference 60.

Line 133-134: During the barley growing cycle, average temperatures range from 8.0 0C (March) to  20.1 0C (June). It needed citation/reference, without data source and citation how the readers knows its reliability.

Answer

We have added reference 61.

Figure 1: Please add global map in the left top side and highlight country with name. In this way, readers will clearly noticed the geographical area of the study site.  

Answer

We have added the name of Spain in the figure.

4. Conclusions

The conclusions of the manuscript little lengthy. I suggest to present only concise conclusion within a paragraph.

Answer

Although we could not present the conclusions in one paragraph we try to reduce them.

Best Regards,

This manuscript is a resubmission of an earlier submission. The following is a list of the peer review reports and author responses from that submission.

Round 1

Reviewer 1 Report

The manuscript describes the results of a field study to investigate how remotely sensed spectral indices, along with other canopy and soil properties, might be used to estimate soil carbon respiration.  The study is limited to a single crop type, but does consider the crop at multiple phenological stages. The methods used are straightfoward and seem to be applied correctly.  The major concern for me is that the study really doesn't break much ground from a remote sensing research perspective, and might be better suited to a specialty agricultural journal.  This is not a comment on the quality of the research, which is adequate to warrant a publication.  It's more an issue with whether the incremental nature of the work is suitable for a journal of remote sensing. 

In addition to the overall suitability of the paper for the journal, the methodological description is a bit sparse in places.  In particular, the various indices are not well-described.  I would suggest that the table 2 be expanded to provide a brief description of each of the indices, emphasizing its typical use in remote sensing analysis.  This would greatly assist the reader in intepreting the relationship between the spectral response of the canopy and carbon respiration.

Reviewer 2 Report

Review of remotesensing-822991

This article focuses on deriving relationships between soil CO2 efflux and spectral indices in rainfed barley crop. The study includes field data collected on a barley field in Spain during 2012 and 2013, which has homogeneous soil, slope, drainage and lighting. The field has been continuously cultivated for decades. The paper uses observations of soil CO2 efflux from an automated soil gas analyzer which was placed at a depth of 5-6 cm in the soil during the growing season. Barley was a good choice given the short canopy and highly uniform plant expression across the fairly small area that was instrumented. Ground spectroradiometer observations were taken to take hyperspectral measurements from 350-2500nm every 15 days when clear sky conditions were available. These measurements were used to calculate seven different spectral indices taken from the literature. In addition, LAI was calculated using LAI measurements across six different points in the field and then averaged to create one plot value per time period. All measurements were compared to a nearby ‘bare soil’ plot without vegetation as a comparison.

I like this paper and thought that it provided some very useful ground information and assessment of the utility of indices at different parts of the growing season. The analysis does highlight the differences between direct observation-based spectral indices and LAI, which is usually a modeled output although here it is a direct observation.

To improve the paper, I would suggest adding to the discussion some information about how scalable the result is. Would the authors suggest, for example, how to transform Sentinel 2a/b data in other fields cultivating barley into Rs observations? Although the model accuracy is fairly low overall, it would be interesting to determine if the authors think lower resolution (10m vs centimeters) would be an issue, along with the inevitable mixed pixels and atmospheric effects. The Remote Sensing journal typically is read by scientists using remote information from satellites, so more effort in connecting these very interesting ground data to those taken remotely would be useful. More effort connecting these results to the interests of the broader remote sensing community would enhance its utility significantly. 

Reviewer 3 Report

This manuscript describes an experiment which attempts to assess the potential to use simple spectral indices to estimate soil respiration and compare these with an estimate using just LAI. This has been done a number of times previously and the authors provide an extremely extensive literature review which confirms this. They justify their study by the use of a different ecosystem from previous publications (dryland barley), this may well be true, but prior publications also included a much larger data set, replication (usually) and a far more sophisticated analysis. In my opinion the data and analysis presented don’t justify a new publication.

That said, the experimental data set seems to have been collected in a rigorous manner and there is much more that can be done with that data. For example the full NIR-SWIR spectral range was collected and this is something not typically available from satellite data; a model of soil respiration in the three ‘sites’ (plot 1, plot 2 and bare soil) using this data, whether by PLS or a machine learning method and comparison with the simulated MODIS data, would be useful. A comparison between the simple linear regression model used and some of the previously published models of soil respiration using MODIS data might be worthwhile. An attempt to split the soil respiration estimate into autotrophic (presumably related to plant biomass, estimated via LAI or VI) and heterotrophic pools might be interesting, although I appreciate that the bare soil site will underestimate heterotrophic respiration. There may be other aspects the authors could build on to make the manuscript more valuable to the reader.

When it comes down to it there are only three ‘sites’ and they are not independent as they are all part of one plot and the analysis used depends on the temporal change in soil respiration and plant growth, which of course is not independent either. In addition to the above comments, this makes the conclusions that can be inferred from the statistical analysis provided somewhat limited as well.

Reviewer 4 Report

Overall comment

The paper is well-written and discusses the quickly evolving domain of remote sensing of terrestrial carbon cycle. There is a lack of novelty in this paper, they focused on existing approaches without proving any advantages. Unfortunately, the inclusion of rainfed crops do not highlight the novelty (line 105).  Moreover, the very small study area do not allow to generalize the results.  In my view, it provides some first insights and this should be mentioned in the title. There are also some points to fix, to improve the clarity of the manuscript, as well as some improvements of the English language required, as detailed below. Below some specific issues are mentioned:

Line 40: Please define here C

Line 95: Define here the acronym SSI

Line 178: Is this time frame considered as a representative of the daily mean soil respiration rate in the roi?

Lines 199-200: Please remove the “”

Line 201 I What about other multispectral (Sentinel-2) or hyperspectal (PRISMA, the platform is already in orbit) sensors that might offer large potential, since their bands can be more promising. I suggest to assess the performance of these sensors to your results, since you can evaluate more indices (Clred edge).

Line 235: Numbers up to ten are generally written in full text

Line 255: ...1.9 m2 m-2, while maximum values...

Line 266: In both cases, the minimum..

Line 295: Please use the defined acronym instead of the Spectral Shape Indices

Line 415: The discussion section is very poor. The limitation of this study should discussed in more detail, e.g. few samples, restricted area and only one and also one soil type. Possible comparison with the sensors aforementioned.

Line 456: …contained…

Line 462 By including…

Line 485: The conclusions are too optimistic, please reconsider this chapter accordingly the 'new' discussion and results.

Reviewer 5 Report

This research aimed to assess Rs linked to crop phenology of a rainfed barley crop throughout two seasons based on spectral indices calculated from field spectroscopy data.

Overall, a really interesting and well written research. I suggest the following minor revisions:

  • References: 105 references for an original article are not justifiable. If it was a review...but this is not the case. I carefully read the paper and I don't agree in using all of such references. Please select not more than 50/60 references.
  • Soil classification (M&M section): please classify your soil at Great Group level at least. "The experimental area results from a filling of anthropic origin consisting of arkosic materials with an abundance of artifacts ("human-transported materials", [63])." It's an incorrect way to describe investigated soils. I suggest changing it in the following way ""The experimental area results from a filling of anthropic origin ("human-transported materials", [Capra et al. 2015]) consisting of arkosic materials with an abundance of artifacts. For such reasons it can be classified as ... according to US Soil Taxonomy System [63])." The new proposed reference is the https://link.springer.com/article/10.1007/s11368-015-1110-x
  •